# Association of Serum Oxysterols with Cholesterol Metabolism Markers and Clinical Factors in Patients with Coronary Artery Disease: A Covariance Structure Analysis

**DOI:** 10.3390/nu15132997

**Published:** 2023-06-30

**Authors:** Yusuke Akiyama, Shunsuke Katsuki, Tetsuya Matoba, Yasuhiro Nakano, Susumu Takase, Soichi Nakashiro, Mitsutaka Yamamoto, Yasushi Mukai, Shujiro Inoue, Keiji Oi, Taiki Higo, Masao Takemoto, Nobuhiro Suematsu, Kenichi Eshima, Kenji Miyata, Makoto Usui, Kenji Sadamatsu, Toshiaki Kadokami, Kiyoshi Hironaga, Ikuyo Ichi, Koji Todaka, Junji Kishimoto, Hiroyuki Tsutsui

**Affiliations:** 1Department of Cardiovascular, Respiratory and Geriatric Medicine, Kyushu University Beppu Hospital, Oita 874-0838, Japan; akiyama.yusuke.509@m.kyushu-u.ac.jp; 2Department of Cardiovascular Medicine, Kyushu University Hospital, Fukuoka 812-8582, Japan; 3Department of Cardiovascular Medicine, Saiseikai Fukuoka General Hospital, Fukuoka 810-0001, Japan; 4Department of Cardiovascular Medicine, Harasanshin Hospital, Fukuoka 812-0033, Japan; 5Department of Cardiovascular Medicine, Japanese Red Cross Fukuoka Hospital, Fukuoka 815-0082, Japan; 6Department of Cardiovascular Medicine, National Hospital Organization Kyushu Medical Centre, Fukuoka 810-0065, Japan; 7Wakaba Heart Clinic, Fukuoka 810-0073, Japan; 8Cardiovascular Center, Steel Memorial Yahata Hospital, Fukuoka 805-8508, Japan; 9Matsuguchi Internal Medicine and Cardiology Clinic, Fukuoka 814-0133, Japan; 10Department of Cardiovascular Medicine, Japan Community Health Care Organization, Kyushu Hospital, Fukuoka 806-8501, Japan; 11Department of Cardiovascular Medicine, Hamanomachi Hospital, Fukuoka 810-0072, Japan; 12Department of Cardiovascular Medicine, Omuta City Hospital, Fukuoka 836-0861, Japan; 13Department of Cardiovascular Medicine, Saiseikai Futsukaichi Hospital, Fukuoka 818-8516, Japan; 14Department of Cardiovascular Medicine, Fukuoka City Hospital, Fukuoka 812-0046, Japan; 15Graduate School of Humanities and Science, Ochanomizu University, Tokyo 112-8610, Japan; 16Center for Clinical and Translational Research, Kyushu University Hospital, Fukuoka 812-8582, Japan; 17School of Medicine and Graduate School, International University of Health and Welfare, Fukuoka 831-8501, Japan

**Keywords:** oxysterol, cardiovascular disease, cholesterol metabolism

## Abstract

Oxysterols have been implicated in the pathogenesis of cardiovascular diseases. Serum levels of oxysterols could be positively correlated with cholesterol absorption and synthesis. However, physiological regulation of various serum oxysterols is largely unknown. The aim of this study was to investigate the relationship between clinical factors and cholesterol metabolism markers, and identify oxysterols associated with cholesterol absorption and synthesis in patients with coronary artery disease. Subjects (*n* = 207) who underwent coronary stenting between 2011 and 2013 were studied cross-sectionally. We measured lipid profiles including serum oxysterols. As for the serum biomarkers of cholesterol synthesis and absorption, oxysterol levels were positively correlated with campesterol and lathosterol. Covariance structure analysis revealed that dyslipidemia and statin usage had a positive correlation with “cholesterol absorption”. Statin usage also had a positive correlation with “cholesterol synthesis”. Several oxysterols associated with cholesterol absorption and/or synthesis. In conclusion, we elucidated the potential clinical factors that may affect cholesterol metabolism, and the associations between various oxysterols with cholesterol absorption and/or synthesis in patients with coronary artery disease.

## 1. Introduction

Oxysterols are oxygenated derivatives of cholesterol with a multitude of biological activities [1,2]. These sterols are detected in the serum at very low concentrations compared to cholesterol [2]. Concerning cardiovascular diseases (CVD), they have been found enriched in atherosclerotic lesion, and their roles in the etiology of cardiovascular diseases have been studied [2]. Furthermore, preclinical and clinical studies suggest that oxysterols are implicated in the pathogenesis of many health conditions, including heart failure, liver disease [3], neurodegenerative disease [4], eye disorder [5], cancer [6], diabetes [7], and infectious disease [8]. They can be produced endogenously by autoxidation and/or enzymatic reactions [1], or provided by food.

Apart from serum lipid levels, alterations in cholesterol metabolism may affect the risk of CVD [9]. Cholesterol in diet is absorbed from the intestine through Niemann-pick C1-like 1, an intestinal cholesterol transporter [10], which is blocked by ezetimibe. In clinical settings, cholesterol absorption can be assessed by measuring the serum level of plant sterols such as campesterol [11], which is not synthesized in humans. In addition to dietary cholesterol intake, the liver synthesizes cholesterol. Cholesterol synthesis can be assessed by measuring serum levels of lathosterol, a precursor of cholesterol [11]. Previous studies have indicated that various clinical factors, such as age, sex, obesity, hypertension, diabetes, renal dysfunction, and lipid-lowering medications, are associated with serum sterol levels [12,13,14,15,16]. Understanding the impact of these clinical factors on cholesterol metabolism and oxysterol levels in patients with CAD is of great importance.

Previous studies showed that high cholesterol absorbers had higher CVD risk [17,18]. We recently conducted the CuVIC trial (Effect of Cholesterol Absorption Inhibior Usage on Target Vessel Dysfunction After Coronary Stenting), in which ezetimibe in combination with statin ameliorated coronary endothelial dysfunction (CED) associated with reduced oxysterol levels in patients with coronary artery disease (CAD) after coronary stenting [19]. We demonstrated that: (1) the incidence of coronary endothelial dysfunction associated with not only serum low-density lipoprotein cholesterol (LDL-C) levels, but also total oxysterol levels; and (2) ezetimibe in combination with statin ameliorated serum levels of campesterol and total oxysterol compared to statin monotherapy. On the other hand, serum levels of several oxysterols, including 24S-hydroxycholesterol and 27-hydroxycholesterol, were reported to positively correlate with markers of cholesterol synthesis, such as serum levels of lathosterol and desmosterol [20]. Another study reported that treatment with statin monotherapy reduced serum levels of oxysterols and decreased cholesterol synthesis markers [21]. These studies suggested that serum levels of oxysterols could be positively correlated with cholesterol absorption and synthesis markers. However, the comprehensive associations of individual oxysterols with either cholesterol absorption or synthesis, or both, are largely unknown, especially among patients with CAD. Moreover, the clinical factors that may affect the cholesterol metabolism and serum oxysterols remain to be elucidated.

In this present study, a sub-analysis of the CuVIC trial, we employed a covariance structure analysis to explore the associations between clinical factors with cholesterol absorption and synthesis, and their associations with serum oxysterols in the patients with CAD.

## 2. Materials and Methods

### 2.1. Trial Design

This current study is a post hoc analysis of the CuVIC trial [19]. The protocol and the main results have been described previously. The Institutional Review Board at Kyushu University approved the protocol and all subjects provided informed consent. The CuVIC trial was a randomized trial of statin monotherapy vs. ezetimibe 10 mg/day + statin combination therapy in 260 patients with CAD who underwent coronary stenting at 11 cardiovascular centers in the period 2011–2013. In both treatment groups, the target LDL-C value was established as 100 mg/dL or lower. Participating physicians were allowed to titrate statin doses to attain the target goal. The primary endpoint of the main study was defined as target vessel dysfunction, which is the composite of target vessel failure occurring within the 6–8 months follow-up period and CED determined by intracoronary injection of acetylcholine at the follow-up coronary angiography (CAG). Exclusion criteria consisted of the following: a scheduled coronary revascularization procedure, participation in concurrent clinical trial that could interfere with this study, end-stage renal failure and on hemodialysis, liver cirrhosis, severe left ventricular dysfunction (ejection fraction < 30%), or contraindication to statins or ezetimibe. Among those patients, we enrolled 207 patients for whom serum oxysterols data measured at baseline were available in this sub-study.

### 2.2. Biomarker Assessment

At the baseline of the CuVIC Trial, blood samples were obtained and LDL-C levels were determined using the Friedewald equation. Alongside routine laboratory tests, including lipid profiling at each participating center, samples were sent to Medical and Biological Laboratories, Co., LTD., Nagoya, Japan. At this facility, measurements were performed to assess levels of apolipoprotein A1 (Apo A1), apolipoprotein B (Apo B), high-sensitive C-reactive protein (hs-CRP), and malondialdehyde modified LDL (MDA-LDL). The non-cholesterol sterols (campesterol, sitosterol, and lathosterol) were measured at SRL, Inc., Tokyo, Japan, using gas chromatography (GC-2010, Shimadzu, Co., Kyoto, Japan). Oxysterols were measured using gas chromatography mass spectrometry (GC/MS QP2010; Shimadzu Co.) equipped with an SPB-1 fused silica capillary column (60 m × 0.25 mm, 0.25 μm phase thickness; Supelco Inc., Bellefonte, PA, USA). Participants provided information regarding their general demographic and health information at the time of enrollment, including sex, age, height, weight, smoking status, history of metabolic syndrome, hypertension, diabetes, dyslipidemia, and current use of medications such as statins, antihypertensive drugs, diabetes drugs, and insulin.

### 2.3. Statistical Analysis

Continuous variables are reported as mean values ± standard deviation or median values with 25th and 75th percentiles depending on their distribution, and categorical variables are reported as numbers and percentages. Categorical variables were compared using a χ^2^ test, and variables were compared using a Student’s *t*-test. The correlation between two factors is expressed as Spearman’s correlation. A *p*-value of less than 0.05 was regarded as statistically significant. Structural equation modeling (SEM) was used to estimate the association between the serum levels of oxysterols, cholesterol synthesis/absorption, and clinical factors. Firstly, we divided subjects into two groups based on the median value of campesterol, and a Student’s *t*-test was performed. We conducted a similar analysis using lathosterol. The purpose of creating a path diagram was to verify the following two hypotheses: firstly, various oxysterols are regulated by two latent variables, “cholesterol absorption” and “cholesterol synthesis”; and secondly, these two latent variables are influenced by several clinical factors. We performed covariance structure analysis using the latent variables for oxysterols. To evaluate the method fit, we used the Comparative Fit Index (CFI) and the Root Mean Square Error of Approximation (RMSEA). Statistical analysis was performed using the JMP software (SAS Institute Inc., Cary, NC, USA). The SEM analyses were performed using a statistical software package IBM SPSS AMOS version 25 (Amos Development Corporation, Meadville, PA, USA).

## 3. Results

### 3.1. Study Population

All subjects underwent successful percutaneous coronary intervention (PCI). The clinical and biochemical data suggest that this cohort showed normal body weight and BMI. Median [IQR] concentration of total cholesterol and LDL-C was 162 [139, 196] mg/dL and 92 [73, 118] mg/dL, respectively. Median concentrations of triglyceride and high-density lipoprotein cholesterol (HDL-C) were within the reference range. Statin was prescribed to 131 patients (62%). Median concentration of campesterol, sitosterol and lathosterol was 3.5 [2.7, 4.7] μg/mL, 1.7 [1.3, 2.5] μg/dL, and 1.0 [1, 1.2] μg/dL, respectively. The prevalence of metabolic syndrome, hypertension, diabetes, dyslipidemia, and ACS at index PCI was 36%, 71%, 47%, 89%, and 38%, respectively (Table 1).

### 3.2. Analysis of Oxysterols in the Study Population

We examined serum levels of oxysterols in the study population. The value of the serum total oxysterol was 1507 [1174–2036] ng/mL. The 7-ketocholesterol values (323 [215–582] ng/mL) accounted for the majority of dietary oxysterols, which include β-epoxycholesterol (β-EPOXY), cholestan-3β,5α,6β-triol (TRIOL), 7β-hydroxycholesterol, and 7-ketocholesterol. The 27-hydroxycholesterol values (401 [323–489] ng/mL) accounted for the majority of intrinsic oxysterols, which include 4β-hydroxycholesterol, 24-hydroxycholesterol, and 27-hydroxycholesterol. The 7α-hydroxycholesterol values (152 [106–267] ng/mL) accounted for the majority of dietary and intrinsic oxysterols, which include 7α-hydroxycholesterol and 25-hydroxycholesterol (Table 1). The histogram of concentration of total oxysterol showed modestly right skewed distributions (Figure 1).

### 3.3. Association of Oxysterols with Cholesterol Absorption and Synthesis

Previous studies suggested that oxysterols in peripheral blood could be positively correlated with cholesterol absorption and synthesis [20,21]. We then investigated the association between the serum level of total oxysterol and cholesterol metabolism biomarkers. The serum level of total oxysterol was positively associated with the cholesterol metabolism biomarkers such as campesterol (spearman R^2^ = 0.021, *p* = 0.035) and lathosterol (spearman R^2^ = 0.06, *p* < 0.001) (Figure 2).

### 3.4. Association of Cholesterol Absorption and Synthesis with Clinical and Biochemical Characteristics

To investigate the oxysterols that correlate with cholesterol absorption and/or synthesis markers, we divided this study population into two subgroups based on the median value of campesterol (3.5 μg/mL) and lathosterol (1.0 μg/mL), respectively. Total cholesterol, HDL-C, and LDL-C were significantly higher in the high campesterol group than the low campesterol group in lipid profiles. Also, total oxysterol, TRIOL, 4β-hydroxycholesterol, 7α-hydroxycholesterol, 7β-hydroxycholesterol, 7-ketocholesterol, and 24-hydroxycholesterol were significantly higher in the high campesterol group than low campesterol group (Table 2). By contrast, triglyceride and MDA-LDL were significantly higher in the high lathosterol group than low lathosterol group in lipid profiles. Total oxysterol, β-EPOX, 7α-hydroxycholesterol, 7β-hydroxycholesterol, 7-ketocholesterol, 24-hydroxycholesterol, 25-hydroxycholesterol, and 27-hydroxycholesterol were significantly higher in the high lathosterol group than low lathosterol group (Table 3).

### 3.5. Path Model Estimated Regression Weights in Oxysterols Revealed Potential Clinical Factors Affecting Cholesterol Metabolism Markers

The multivariate analyses would not be enough to identify clinical factors that predict the patients with high cholesterol absorption or synthesis because the factors confound each other, and it is challenging to make up the characteristics of each clinical factors explicit using the respective equation model. We then employed the proposed path model 1 to compensate the confounding factors (Figure 3). In brief, the clinical factors potentially confound each other; the association between two factors is linked by the bidirectional arrows. Statistically insignificant paths are drawn as thin arrows. Paths between variables are drawn from independent to dependent variables, with a directional arrow for every regression model—namely, from age, sex, hypertension, diabetes, dyslipidemia, smoking, statin usage, and hs-CRP to the latent variable “cholesterol absorption” and to the other latent variable “cholesterol synthesis”. Furthermore, directional arrows were drawn between each latent variable and each oxysterol positively correlated with cholesterol metabolism markers (Table 2 and Table 3). The proposed path model 1 revealed dyslipidemia (β [standardized regression coefficient]:0.163, *p* = 0.028) and statin usage (β:0.175, *p* = 0.017) showed a positive correlation with the latent variable “cholesterol absorption”, which shows a significant correlation with campesterol (β:0.273, *p* < 0.001). Consistent with the ratio of male to female patients with dyslipidemia in Japan [17], dyslipidemia negatively correlated with male sex (β:−0.14, *p* = 0.041). Also, it positively correlated with high sensitivity CRP (β:0.19, *p* = 0.016).

On the other hand, statin usage (β:0.261, *p* < 0.001) showed a positive correlation with the other latent variable “cholesterol synthesis”, which shows a significant correlation with lathosterol (β:0.312, *p* < 0.001). (Figure 3, Appendix A).

### 3.6. Path Model Revealed the Associations between Individual Oxysterols and Latent Cholesterol Absorption and Synthesis

The proposed path model 1 also revealed the associations between individual oxysterols and latent variables of cholesterol absorption and synthesis. Among the oxysterols, TRIOL (β:0.542, *p* < 0.001) and 4β-hydroxycholesterol (β:0.958, *p* < 0.001) were positively regulated by “cholesterol absorption”. 7-ketocholesterol (β:0.897, *p* < 0.001), 27-hydroxycholesterol (β:0.236, *p* < 0.001) and β-EPOX (β:0.678, *p* < 0.001) were positively regulated by “cholesterol synthesis”. 24-hydroxycholesterol (β:0.153, *p* = 0.019 and β:0.609, *p* < 0.001), 25-hydroxycholesterol (β:0.355, *p* < 0.001 and β:0.665, *p* < 0.001), 7α-hydroxycholesterol (β:0.196, *p* < 0.002 and β:0.72, *p* < 0.001) and 7β-hydroxycholesterol (β:0.277, *p* < 0.001 and β:0.775, *p* < 0.001) were positively regulated by both cholesterol absorption and synthesis. Based on the result of this path analysis, it was noted that oxysterols may be regulated via distinct mechanisms.

Furthermore, we employed an alternative proposed path model 2 to examine the impact of other clinical factors including BMI, BNP, and ACS at index PCI (Appendix A). When constructing path model 2, there were limitations on the number of variables that could be included in the path model. As a result, variables such as gender, smoking, and hs-CRP, which did not show significant correlations with the latent variable “cholesterol absorption” and “cholesterol synthesis” in path model 1, were excluded from the model. Instead, BMI, BNP, and ACS at index PCI were added as additional variables to the path model 2. None of these additional clinical factors showed significant correlations with both latent variables of cholesterol absorption and synthesis. In this proposed path model, BMI showed a positive correlation with hypertension (β:0.323, *p* = 0.004), diabetes (β:0.247, *p* = 0.044), and dyslipidemia (β:0.18, *p* = 0.018). This is consistent with previous reports indicating a high likelihood of comorbidity between obesity and these disease [22,23]. Moreover, BNP showed a positive correlation with age (β:0.386, *p* < 0.001) and ACS at index PCI (β:0.129, *p* = 0.004), both of which are well-established risk factors for heart failure [24]. The associations between individual oxysterols and latent variables in the path model 2 were consistent with those observed in path model 1.

## 4. Discussion

In the present study, we investigated the potential clinical factors that may affect cholesterol metabolism, and the association between various oxysterols with cholesterol absorption and/or synthesis in the patients with coronary artery disease. The key findings include: (1) serum total oxysterol level showed significant positive correlations with both campesterol and lathosterol, which are markers of cholesterol absorption and synthesis; (2) the proposed path model revealed the latent variable “cholesterol absorption” that was associated with dyslipidemia and statin usage positively regulated campesterol and the latent variable “cholesterol synthesis”, which is associated with statin usage positively regulated lathosterol; and (3) among the oxysterols, TRIOL and 4β-hydroxycholesterol were positively regulated by the latent “cholesterol absorption”. 7-ketocholesterol, 27-hydroxycholesterol and β-EPOX were positively regulated by the latent “cholesterol synthesis”. 24-hydroxycholesterol, 25-hydroxycholesterol, 7α-hydroxycholesterol, and 7β-hydroxycholesterol were positively regulated by both of these latent variables.

Most of the oxysterol values in our study in patients with CAD were found to be higher than the range previously reported for different healthy population [21]. Especially the values of 7-ketocholesterol and 27-hydroxycholesterol were more than two-fold higher compared to those reported previously (range for 7-ketocholesterol 10.7–98.0 ng/mL and 27-hydroxycholesterol 43.6–196.0 ng/mL, respectively). In contrast, 24-hydroxycholesterol and 25-hydroxycholesterol values in this population were within the previously reported range [21]. The 7-ketocholesterol is one of the most common dietary oxysterols and abundantly presented in human atherosclerotic lesions [2]. We previously demonstrated in animal studies that 7-ketocholesterol possesses proinflammatory properties in vascular cells, such as promoting endothelial cell proliferation and tissue factor expression in smooth muscle cells [25]. Furthermore, 7-ketocholesterol induce monocyte/macrophage mediated inflammation in myocardial infarction by inducing endoplasmic reticulum (ER) stress [26]. Umetani et al. found that 27-hydroxycholesterol promoted atherosclerosis via proinflammatory processes mediated by estrogen receptor alpha (ERα), and this oxysterol attenuated estrogen-related atheroprotection [27]. 27-hydroxycholesterol, the major oxysterol in serum and atheroma lesions, is generated enzymatically from cholesterol in several tissues, including atheroma lesions in direct relation to serum levels [2]. This study included CAD patients who are assumed more prevalent to progressive atherosclerosis. In this respect, a possible mechanism by which advanced level of systemic atherosclerosis might cause the elevation of serum oxysterols level.

Stiles et al. reported that circulating 4β-hydroxycholesterol could be influenced by diet based on an observed positive correlation with serum concentrations of plant sterols [28]. Consistently, serum level of total oxysterol was correlated with both of campesterol and lathosterol. We further found that campesterol and lathosterol were correlated with individual levels of different lipids and oxysterols, suggesting potential shared metabolic pathways. In general, circulating oxysterols are positively correlated with serum total cholesterol, triglycerides, and LDL- cholesterol, and negatively correlated with HDL-cholesterol [28]. Most serum oxysterols are found in the LDL-C and HDL-C fractions [29], suggesting that oxysterols are transported in serum with cholesterols. Common origins may explain numerous positive correlations, such as those between most sterols and cholesterols. These correlations may reflect cotransport in serum lipoprotein particles [29]. Additionally, positive correlations can be observed between sterols, such as sitosterol and campesterol, which can be attributed to their dietary origin, as well as their absorption and excretion via ABCG5/ABCG8 [30].

We elucidated the associations between clinical factors and cholesterol metabolism, and then those between cholesterol metabolism and various oxysterols using the cutting-edge path model. To the best of our knowledge, there were no reports that evaluated these associations using a covariance structure analysis. In this analysis, statin usage not only positively correlated with “cholesterol absorption”, but also with “cholesterol synthesis”, contrary to some previous reports that demonstrated reductions in circulating oxysterols even with short-term statin treatment [31,32]. The reason for this discrepancy could be that the subjects in this study were hypercholesterolemic patients with elevated cholesterol synthesis, which led to a statin usage. Dyslipidemia, which was positively correlated with cholesterol absorption, was also positively correlated with hs-CRP. Emerging evidence suggests that serum oxysterols may associate with systemic inflammation through interaction with receptors other than the LXRs [33,34]. However, we did not find that the hs-CRP as an inflammatory marker associated with serum oxysterols level in this path model although hs-CRP had a positive correlation with dyslipidemia that was positively correlated with “cholesterol absorption”. This may be due to the fact that most of the study population were taking statins. In addition, there is inconsistency as for the link between diabetes and cholesterol absorption in previous studies [35,36]. Matsumura et al. reported that diabetes associated with lower cholesterol absorption independently [14]. However, we did not find a correlation between diabetes and cholesterol absorption or synthesis in this study. This suggested that the subjects receiving antidiabetic drugs may have their circulating glucose levels controlled to targets established for the prevention of cardiovascular disease. The path analysis revealed that oxysterols were differently regulated by “cholesterol absorption” alone, “cholesterol synthesis” alone, or both. Oxysterols can be classified based on its origin. Some of the endogenous oxysterols are produced by organ-specific enzyme (i.e., 24-hydroxycholesterol by brain-specific CYP46A1, and 27-hydroxycholesterol by liver-specific CYP27A1) [21]. On the other hand, 7β-hydroxycholesterol, 7-ketocholesterol, and β-EPOX can be formed by non-enzymatic reaction, i.e., auto-oxidation [37]. Furthermore, 25-hydroxycholesterol and 7α-hydroxycholesterol can also be formed by autooxidation pathways. In the path analysis of this study, the observed correlations for some oxysterols such as 27-hydroxycholesterol and “cholesterol synthesis” were consistent with known metabolic pathways. On the other hand, 7-ketocholesterol, a dominant non-enzymatically derived oxysterol, showed a significant correlation with “cholesterol synthesis” rather than “cholesterol absorption”. This notion agrees with our animal study showing that high cholesterol diet and infusion of angiotensin II that induces oxidative stress increases serum 7-ketocholesterol, which are inhibited by rosuvastatin or ezetimibe monotherapy [25]. The path analysis revealed that oxysterols are intricately regulated in vivo through various biological processes and oxidative stress mechanisms beyond their origins.

It is necessary to be aware that the cholesterol metabolism affects the serum oxysterols, some of which are atherogenic. We have previously shown that dietary oxysterols accelerate atherosclerotic plaque destabilization in hypercholesterolemic mice, and this acceleration is ameliorated by ezetimibe monotherapy, which is associated with decreases in serum oxysterol levels [38]. These findings support the notion that ezetimibe, a cholesterol absorption inhibitor, may possess beneficial anti-atherogenic effects through lowering oxysterols effectively by the different mechanisms with statins. Addition of ezetimibe to statin monotherapy may be a plausible therapeutic strategy to lower oxysterols that positively regulated by both “cholesterol absorption” and “cholesterol synthesis” in the path model. Furthermore, statin and ezetimibe combination therapy ameliorated endothelial dysfunction compared with the statin monotherapy, and this effect was associated with larger decrease in the levels of atherogenic oxysterols. The CuVIC main trial did not show a direct correlation between cholesterol metabolism markers and endothelial function; however, from the results of this sub study, we found that cholesterol metabolism may regulate oxysterol levels. Based on these findings, cholesterol metabolism markers might predict endothelial function even though there have been no studies demonstrating this [39].

Frailty is increasingly recognized in the field of heart failure [40]. This is not only because heart failure and frailty share aging as a predisposing factor but also because both conditions are strongly associated with systemic multisystem dysfunction. As the participants of this study were relatively young, we anticipate that there was a low prevalence of frailty. Since age and BMI are reported to be associated with frailty [41], we further analyzed the association between cholesterol metabolism markers between each of age and BMI. In this study, the path analysis revealed that neither age nor BMI was associated with cholesterol absorption/synthesis. The prevalence of heart failure is rising globally [42], and it would be beneficial if we could prevent the heart failure through therapeutic interventions targeting cholesterol metabolism and oxysterols. Songs et al. reported that high 7-ketocholesterol levels are associated with increased risk of cardiovascular events including heart failure [43]. Furthermore, cholesterol metabolism markers also have been reported as predictive factors for heart failure [44]. In this study, heart failure was not included as an endpoint, and no data related to the incidence of heart failure were collected. Although it is difficult to show a direct correlation between heart failure and each of sterols and oxysterols, factors known to be associated with heart failure, including age, gender, BNP level, and comorbidity such as hypertension and diabetes, are not directly correlated with cholesterol metabolism in this study. Interventions targeting oxysterols may have the potential to reduce the risk of heart failure, but further studies are needed.

There are several limitations in this study. Firstly, this study is a cross-sectional study, and longitudinal studies over a long period of time remain to be performed in the future. Secondly, the CuVIC population was composed of data from patients with coronary diseases and various comorbidities and medications. Therefore, the population differed from the common population. On the other hand, the relatively large sample size was one of the strengths of this study which allowed us to analyze and evaluate the association between oxysterols and clinical factors. Thirdly, Statin medications in the study population may have influenced the results of the path analysis. None of the patients in this sub study cohort were receiving ezetimibe at the time of study enrollment. Hence, we were unable to assess the effect of ezetimibe on baseline sterol levels. Finally, we did not collect sufficient data related to frailty or the incidence of heart failure, making it difficult to add information in this regard.

## 5. Conclusions

This study showed the potential clinical factors that may affect cholesterol metabolism, and the association between various oxysterols with cholesterol absorption and/or synthesis using the samples of the CuVIC study. These results might lead to a precision medicine in which we can apply patients with statin or ezetimibe monotherapy or combination of statin and ezetimibe based on individual oxysterol profiles. Further studies are needed to clarify the mechanisms underlying these associations.

## Figures and Tables

**Figure 1 nutrients-15-02997-f001:**
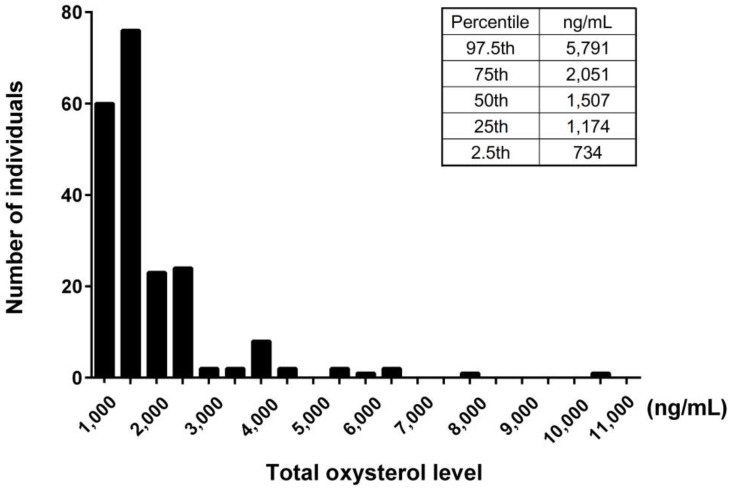
Histogram of serum total oxysterol. The distribution of serum total oxysterol level was shown by density. The 2.5th, 25th, 50th, and 97.5th percentile levels were also indicated.

**Figure 2 nutrients-15-02997-f002:**
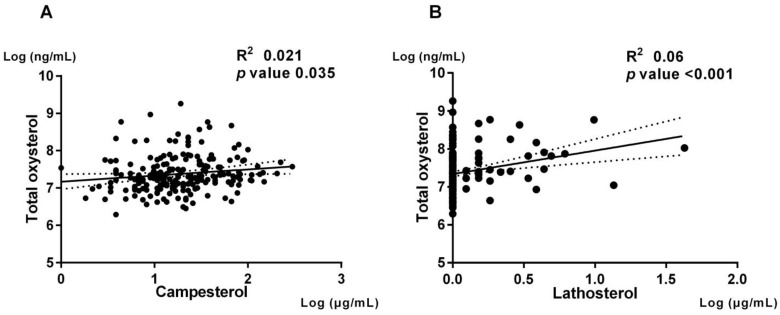
Correlation between total oxysterol level and cholesterol metabolism markers. The correlation of total oxysterol level and cholesterol metabolism biomarkers were shown. Panel (**A**) (Left) shows regression line of total oxysterol and campesterol. Panel (**B**) (Right) shows regression line of total oxysterol and lathosterol.

**Figure 3 nutrients-15-02997-f003:**
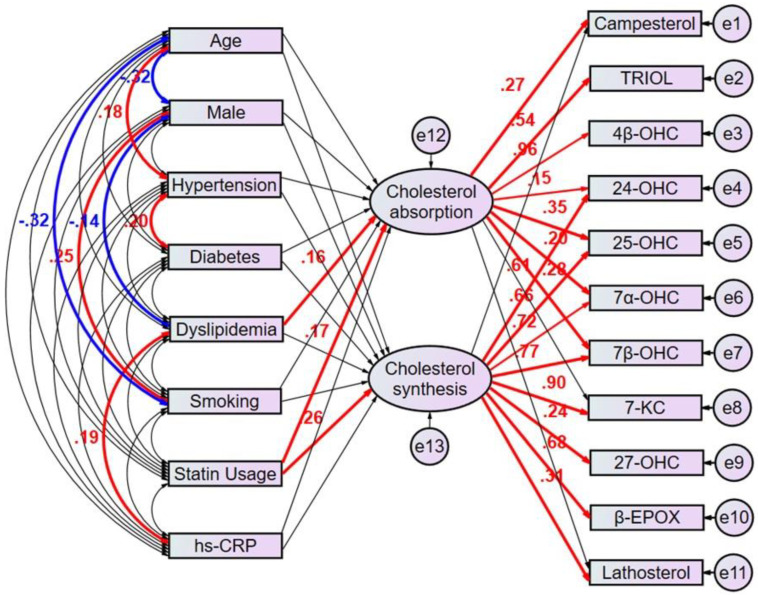
Proposed Path model 1. This path has a coefficient showing the standardized coefficient of regressing independent variables on the dependent variable of the relevant path. These variables indicate standardized regression coefficients (direct effect) [bold capitals]. A two-way arrow between two variables indicates a correlation between those two variables. The total variance in a de-pendent variable for every regression is theorized to be caused by either independent variables of the model or extra-neous variables. hs-CRP = high sensitivity C-reactive protein, TRIOL = cholestan-3β,5α,6β-triol, 4β-OH-C = 4β-hydroxy-cholesterol, 24-OH-C = 24-hydroxy-cholesterol 7-keto-C = 7-keto-cholesterol, β-epoxy-C = β-epoxy-cholesterol.

**Table 1 nutrients-15-02997-t001:** Main clinical and biochemical characteristics of the whole study population.

Variable	Whole Cohort
Sex Male	163 (78%)
Age (years)	66 [61–78]
Weight (kg)	63.3 [57.6–73.3]
BMI (kg/m^2^)	24.3 [22–26.7]
Metabolic syndrome	74 (36%)
Hypertension	147 (71%)
Diabetes	98 (47%)
Dyslipidemia	186 (89%)
Family History	38 (18%)
Smoking	65 (31%)
ACS at index PCI	79 (38%)
T-Chol (mg/dL)	162 [139.5–196]
TG (mg/dL)	122 [90–168]
HDL-C (mg/dL)	43 [35–51]
LDL-C (mg/dL)	92 [73–118]
apoA1 (mg/dL)	116 [106–128]
apoB (mg/dL)	69 [60–83]
Campesterol (μg/mL)	3.5 [2.7–4.7]
Sitosterol (μg/mL)	1.7 [1.3–2.5]
Lathosterol (μg/mL)	1 [1–1.2]
MDA-LDL (U/L)	66 [51–85]
hs-CRP (mg/dL)	0.24 [0.09–0.49]
BNP (pg/mL)	52.3 [18.9–118.3]
Uric acid (mg/dL)	5.7 [4.9–6.6]
Creatinine (mg/dL)	0.82 [0.7–0.98]
Glucose (mg/dL)	110 [95–133]
HbA1C (%)	5.9 [5.4–6.7]
β-Blocker	126 (61%)
ACE inhibitor	63 (30%)
ARB	80 (38%)
CCB	87 (42%)
Nitrate	42 (20%)
Antidiabetics	71 (34%)
Insulin	19 (9%)
Statins before entry	131 (62%)
Dietary oxysterol (ng/mL)	738 [513–1121]
β-epoxy-C (ng/mL)	121 [75–178]
β-TRIOL (ng/mL)	119 [85–175]
7β-OH-C (ng/mL)	101 [65–203]
7-keto-C (ng/mL)	323 [215–582]
Intrinsic oxysterol (ng/mL)	522 [434–664]
4β-OH-C (ng/mL)	91 [61–125]
22R-OH-C (ng/mL)	0.31 [0–0.04]
24-OH-C (ng/mL)	21 [14–36]
27-OH-C (ng/mL)	401 [323–489]
Dietary and intrinsic (ng/mL)	174 [122–299]
7α-OH-C (ng/mL)	152 [106–267]
25-OH-C (ng/mL)	22 [13–39]
Total oxysterol (ng/mL)	1507 [1174–2036]

Data are expressed as medians and interquartile ranges for continuous variables and as percentages for categorical variables. BMI = body mass index, ACS = Acute coronary syndrome, PCI = percutaneous coronary intervention, T-Chol = total cholesterol, TG = triglyceride, HDL-C = high-density lipoprotein cholesterol, LDL-C = low-density cholesterol, apoA1 = apolipoprotein A1, apoB = apolipoprotein B, MDA-LDL = malondialdehyde-modified LDL, hs-CRP = high sensitivity C-reactive protein, HbA1C = hemoglobin A1C, ACE inhibitor = angiotensin-converting enzyme inhibitor, ARB = angiotensin II receptor blocker, CCB = calcium channel blocker, β-epoxy-C = β-epoxy-cholesterol, TRIOL = cholestan-3β,5α,6β-triol, 7-keto-C = 7-keto-cholesterol, 4β-OH-C = 4β-hydroxy-cholesterol, 27-OH-C = 27-hydroxy-cholesterol.

**Table 2 nutrients-15-02997-t002:** Comparison of lipid profiles and oxysterols based on campesterol.

		Campesterol		
		Low (*n* = 107)	High (*n* = 100)	*p* Value
Lipid Profiles				
T-Chol, (mg/dL)		156 ± 39	168 ± 38	0.01
TG, (mg/dL)		129 ± 75	113 ± 75	0.83
HDL-C, (mg/dL)		41 ± 12	45 ± 12	<0.01
LDL-C, (mg/dL)		85 ± 32	98 ± 35	0.049
MDA-LDL, (U/I)		73 ± 29	67 ± 24	0.23
Oxysterols, (ng/mL)	Origin			
Total oxysterol		1422 ± 1071	1569 ± 1400	0.025
β-epoxy-C	N	115 ± 157	130 ± 131	0.67
TRIOL	N	106 ± 138	130 ± 99	0.035
4β-OH-C	E	84 ± 60	96 ± 49	0.046
7α-OH-C	E/N	138 ± 206	158 ± 196	<0.001
7β-OH-C	N	90 ± 139	119 ± 194	0.03
7-keto-C	N	299 ± 581	375 ± 814	0.02
22R-OH-C	E	0.38 ± 1.9	0.22 ± 0.55	0.74
24-OH-C	E	19 ± 42	24 ± 38	0.02
25-OH-C	E/N	19 ± 55	26 ± 63	0.054
27-OH-C	E	389 ± 158	409 ± 140	0.31

Data are expressed as means and SD for continuous variables. T-Chol = total cholesterol, TG = triglyceride, HDL-C = high-density lipoprotein cholesterol, LDL-C = low-density cholesterol, MDA-LDL = malondialdehyde-modified LDL, β-epoxy-C = β-epoxy-cholesterol, TRIOL = cholestan-3β,5α,6β-triol, 7-keto-C = 7-keto-cholesterol, 4β-OH-C = 4β-hydroxy-cholesterol, 22R-OH-C = 22R-hydroxy-cholesterol, E = enzymic origin in humans, N = nonenzymic origin in humans.

**Table 3 nutrients-15-02997-t003:** Comparison of lipid profiles and oxysterols based on lathosterol.

		Lathosterol		
		Low (*n* = 174)	High (*n* = 33)	*p* Value
Lipid Profiles				
T-Chol, (mg/dL)		168 ± 40	178 ± 33	0.19
TG, (mg/dL)		135 ± 71	165 ± 75	0.027
HDL-C, (mg/dL)		44 ± 12	43 ± 11	0.83
LDL-C, (mg/dL)		97 ± 34	100 ± 28	0.56
MDA-LDL, (U/I)		68 ± 26	80 ± 28	0.018
Oxysterols, (ng/mL)	Origin			
Total oxysterol		1710 ± 1142	2528 ± 1556	0.001
β-epoxy-C	N	144 ± 117	218 ± 236	0.007
TRIOL	N	148 ± 117	159 ± 140	0.63
4β-OH-C	E/N	98 ± 54	105 ± 63	0.49
7α-OH-C	E/N	183 ± 180	340 ± 246	<0.001
7β-OH-C	N	149 ± 151	240 ± 231	0.005
7-keto-C	N	494 ± 671	848 ± 831	0.008
22R-OH-C	E	0.31 ± 1.5	0.31 ± 0.9	0.99
24-OH-C	E	31 ± 38	50 ± 45	0.014
25-OH-C	E/N	35 ± 45	69 ± 103	0.027
27-OH-C	E	418 ± 145	489 ± 163	0.013

Data are expressed as means and SD for continuous variables. T-Chol = total cholesterol, TG = triglyceride, HDL-C = high-density lipoprotein cholesterol, LDL-C = low-density cholesterol, MDA-LDL = malondialdehyde-modified LDL, β-epoxy-C = β-epoxy-cholesterol, TRIOL = cholestan-3β,5α,6β-triol, 7-keto-C = 7-keto-cholesterol, 4β-OH-C = 4β-hydroxy-cholesterol, 22R-OH-C = 22R-hydroxy-cholesterol, E = enzymic origin in humans, N = nonenzymic origin in humans.

## Data Availability

All authors share full access to all the data in the study. Yusuke Akiyama and Tetsuya Matoba take responsibility for the integrity of the data and the accuracy of the data analysis.

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
