# Peer review of "Association of Serum Oxysterols with Cholesterol Metabolism Markers and Clinical Factors in Patients with Coronary Artery Disease: A Covariance Structure Analysis"

_nutrients, 2023, doi:10.3390/nu15132997_

Round 1
Reviewer 1 Report
In this study, the authors investigated the associations among serum oxysterols and clinical factors, including cholesterol metabolism markers in patients with CAD. They found significant associations among several of them. I think this study is quite unique, since they assessed serum sterols among patients with CAD where such biomarkers are rarely measured. I would like to raise a couple of comments listed below.
1. It would be great if the authors can show us the associations between serum sterols and endothelial function they assessed in a mail study.
2. Also this reviewer wonders if serum sterols are associated with severity of CAD and/or BNP levels.
3. Did the ezetimibe affect the sterols among them? And how much?
4. Please provide the perspectives relating this issue from the view point of heart failure.
Author Response
We appreciate the thoughtful and constructive comments from the reviewers to our manuscript entitled “Association of Serum Oxysterols with Cholesterol Metabolism Markers and Clinical Factors in Patients with Coronary Artery Disease: A Covariance Structure Analysis.” We extensively revised our manuscript according to the comments from the reviewers. Our point-to-point responses to the comments are described as follows.
Comment #1-1
It would be great if the authors can show us the associations between serum sterols and endothelial function they assessed in a main study.
Response to Comment #1-1
We appreciate this reviewer’s comment. In the main study, the CuVIC trial has shown that ezetimibe administered with statins ameliorated endothelial dysfunction compared with the statin monotherapy, and this effect was associated with larger decreases in the levels of atherogenic oxysterols. In relation to cholesterol metabolism, the levels of the cholesterol absorption markers campesterol and sitosterol significantly increased in the statin monotherapy, whereas levels of these markers decreased in the ezetimibe and statin combination therapy. Although the main study did not show a direct correlation between cholesterol metabolic markers and endothelial function, however, from the results of this sub study, we found that cholesterol metabolism may regulate oxysterol levels, which may cause endothelial dysfunction as shown in the main study. Based on these finding, cholesterol metabolism markers might predict endothelial function even though there have been few studies demonstrating the association between cholesterol metabolism markers and endothelial function. (1)
We added the description that mentions above in Page 11, lines 23- lines 30 in the revised manuscript.
Reference
- Hallikainen M, Lyyra-Laitinen T, Laitinen T, Agren JJ, Pihlajamaki J, Rauramaa R, et al. Endothelial function in hypercholesterolemic subjects: Effects of plant stanol and sterol esters. Atherosclerosis. 2006;188(2):425-32.
Comment #1-2
Also this reviewer wonders if serum sterols are associated with severity of CAD and/or BNP levels.
Response to Comment #1-2
We appreciate this reviewer’s insightful comment. ACS (Acute Coronary Syndrome) patients are speculated to have a higher likelihood of having more severe coronary artery disease (CAD). We conducted additional path analyses as shown in Supplementary Figure to investigate the correlation between the disease pattern (ACS or non-ACS) at index PCI and the level of BNP and cholesterol metabolism and oxysterol level. We addressed the following results in Page 9, lines 15- lines 31 in the revised manuscript.
Furthermore, we employed an alternative proposed path model 2 to examine the impact of other clinical factors including BMI, BNP, and ACS at index PCI (Supplementary Figure, Supplementary table 2). When constructing path model 2, there were limitations on the number of variables that could be included in the path model. As a result, variables such as gender, smoking, and hs-CRP, which did not show significant correlations with cholesterol metabolism in path model 1, were excluded from the model. Instead, BMI, BNP, and ACS at index PCI were added as additional variables to the path model 2. None of these additional clinical factors showed significant correlations with both latent variables of cholesterol absorption and synthesis. In this proposed path model, BMI showed a positive correlation with hypertension (β:0.323, p=0.004), diabetes (β:0.247, p=0.044), and dyslipidemia (β:0.18, p=0.018). This is consistent with previous reports indicating a high likelihood of comorbidity between obesity and these disease (2, 3). Moreover, BNP showed a positive correlation with age (β:0.386, p<0.001) and ACS at index PCI (β:0.129, p=0.004), both of which are well-established risk factors for heart failure (4). The associations between individual oxysterols and latent variables in the path model 2 were consistent with those observed in path model 1.
Reference
- Piche ME, Tchernof A, Despres JP. Obesity Phenotypes, Diabetes, and Cardiovascular Diseases. Circ Res. 2020;126(11):1477-500.
- Franklin SS, Pio JR, Wong ND, Larson MG, Leip EP, Vasan RS, et al. Predictors of new-onset diastolic and systolic hypertension: the Framingham Heart Study. Circulation. 2005;111(9):1121-7.
- Shiba N, Nochioka K, Miura M, Kohno H, Shimokawa H. Trend of westernization of etiology and clinical characteristics of heart failure patients in Japan--first report from the CHART-2 study. Circ J. 2011;75(4):823-33.
Comment #1-3
Did the ezetimibe affect the sterols among them? And how much?
Response to Comment #1-3
We appreciate this reviewer’s comment. None of the patients in this sub study cohort were taking ezetimibe at the study enrollment. Therefore, it was not possible to evaluate the impact of ezetimibe on baseline sterol levels. We mentioned this point as one of the study limitations. Citing the results from the main study of the CuVIC trial, the impact of ezetimibe on serum sterol levels can be described as follows. In relation to cholesterol metabolism markers and ezetimibe, the levels of the campesterol were significantly increased in the statin monotherapy (baseline 4.13±2.27 mg/mL, follow up 4.99±2.33 mg/mL, p<.0001), whereas levels of these markers decreased in the ezetimibe and statin combination therapy (baseline 3.76±1.58 mg/mL, follow up 2.18±1.13 mg/mL, p<.0001). Regarding lathosterol, there were no significant changes in statin monotherapy (baseline 1.10±0.45 mg/mL, follow up 1.10±0.37 mg/mL, p=0.943). However, lathosterol levels modestly increased after treatment in the group of statin and ezetimibe combination therapy, in which the use of moderate- or lower-dose statins was more prevalent than that of statin monotherapy (baseline 1.10±0.29 mg/mL, follow up 1.34±0.54 mg/mL, p<.0001). We have added this explanation to the limitation section (Page 12, lines 4- lines 6 in the revised manuscript) .
Comment #1-4
Please provide the perspectives relating this issue from the view point of heart failure.
Response to Comment #1-4
We appreciate this reviewer’s comment. In this study, heart failure was not included as an endpoint, and data related to heart failure events were not collected. Therefore, it is challenging to make any specific references to the association between sterols and the incidence rate of heart failure events. Songs et al reported that high 7-KC levels are associated with increased risk of cardiovascular events including heart failure.(5) Furthermore, cholesterol metabolism markers also have been reported as predictive factors for cardiovascular events, including heart failure(6).In this sub study, clinical factors such as dyslipidemia and statin usage were shown to potentially regulate oxysterols through cholesterol metabolism. On the other hand, factors known to be associated heart failure, including age, gender, BNP, and comorbidity such as hypertension and diabetes are not directly correlated with cholesterol metabolism in this sub study. Interventions targeting oxysterols may have the potential to reduce the risk of heart failure. We addressed these points in Page 11, lines 38 to line 50 in the revised manuscript.
Reference
- Song J, Wang D, Chen H, Huang X, Zhong Y, Jiang N, et al. Association of Plasma 7-Ketocholesterol With Cardiovascular Outcomes and Total Mortality in Patients With Coronary Artery Disease. Circ Res. 2017;120(10):1622-31.
6. Sawamura A, Okumura T, Hiraiwa H, Aoki S, Kondo T, Ichii T, et al. Cholesterol metabolism as a prognostic marker in patients with mildly symptomatic nonischemic dilated cardiomyopathy. J Cardiol. 2017;69(6):888-94.
Reviewer 2 Report
I found that this article (with same title) is already able to access in Google as a preprint, so I'm not sure about its acceptance. Please find the link below.
https://europepmc.org/article/ppr/ppr669436
https://www.preprints.org/manuscript/202306.0099/v1
Association of Serum Oxysterols with Cholesterol Metabolism Markers and Clinical Factors in Patients with Coronary Artery Disease: A Covariance Structure Analysis
The manuscript entitled “Association of Serum Oxysterols with Cholesterol Metabolism Markers and Clinical Factors in Patients with Coronary Artery Disease: A Covariance Structure Analysis” by Yusuke et al., is an interesting article. Here, the authors measured lipid profiles including serum oxysterols and elucidated the potential clinical factors that may affect cholesterol metabolism, and the associations between various oxysterols with cholesterol absorption and/or synthesis in patients with coronary artery disease
Overall, the information presented in this manuscript is useful and I approve its publication after some minor updates.
Minor comments: I suggest that these comments be updated before publication.
a. Introduction part is good, but there is not much information about the cholesterol metabolism markers as well as the associated clinical factors.
b. It will be good to add about the frailty/BMI parameters of the patients.
I found that this article (with same title) is already able to access in Google as a preprint, so I'm not sure about its acceptance. Please find the link below.
https://europepmc.org/article/ppr/ppr669436
https://www.preprints.org/manuscript/202306.0099/v1
Association of Serum Oxysterols with Cholesterol Metabolism Markers and Clinical Factors in Patients with Coronary Artery Disease: A Covariance Structure Analysis
The manuscript entitled “Association of Serum Oxysterols with Cholesterol Metabolism Markers and Clinical Factors in Patients with Coronary Artery Disease: A Covariance Structure Analysis” by Yusuke et al., is an interesting article. Here, the authors measured lipid profiles including serum oxysterols and elucidated the potential clinical factors that may affect cholesterol metabolism, and the associations between various oxysterols with cholesterol absorption and/or synthesis in patients with coronary artery disease
Overall, the information presented in this manuscript is useful and I approve its publication after some minor updates.
Minor comments: I suggest that these comments be updated before publication.
a. Introduction part is good, but there is not much information about the cholesterol metabolism markers as well as the associated clinical factors.
b. It will be good to add about the frailty/BMI parameters of the patients.
Author Response
We appreciate the thoughtful and constructive comments from the reviewers to our manuscript entitled “Association of Serum Oxysterols with Cholesterol Metabolism Markers and Clinical Factors in Patients with Coronary Artery Disease: A Covariance Structure Analysis.” We extensively revised our manuscript according to the comments from the reviewers. Our point-to-point responses to the comments are described as follows.
Comment #2-1
Introduction part is good, but there is not much information about the cholesterol metabolism markers as well as the associated clinical factors.
Response to Comment #2-1
We appreciate this insightful comment. In this study, we revealed the associations between sterols as cholesterol metabolism markers and several clinical factors. Previous reports have indicated that various clinical factors such as age(7), sex(8), obesity, hypetension, diabetes(9), renal dysfunction(10), and lipid-lowering medications(11) are associated with serum sterol levels. In the Introduction section, we provided additional explanations regarding the clinical factors known from previous reports to be related to sterols. (page 2, line 17-21)
Reference
7 Silbernagel G, Fauler G, Renner W, Landl EM, Hoffmann MM, Winkelmann BR, et al. The relationships of cholesterol metabolism and plasma plant sterols with the severity of coronary artery disease. J Lipid Res. 2009;50(2):334-41.
- Matthan NR, Pencina M, LaRocque JM, Jacques PF, D'Agostino RB, Schaefer EJ, et al. Alterations in cholesterol absorption/synthesis markers characterize Framingham offspring study participants with CHD. J Lipid Res. 2009;50(9):1927-35.
- Matsumura T, Ishigaki Y, Nakagami T, Akiyama Y, Ishibashi Y, Ishida T, et al. Relationship between Diabetes Mellitus and Serum Lathosterol and Campesterol Levels: The CACHE Study DM Analysis. J Atheroscler Thromb. 2022.
- Rogacev KS, Pinsdorf T, Weingärtner O, Gerhart MK, Welzel E, van Bentum K, et al. Cholesterol synthesis, cholesterol absorption, and mortality in hemodialysis patients. Clin J Am Soc Nephrol. 2012;7(6):943-8.
- Miettinen TA, Gylling H. Effect of statins on noncholesterol sterol levels: implications for use of plant stanols and sterols. Am J Cardiol. 2005;96(1A):40D-6D
Comment #2-2
It will be good to add about the frailty/BMI parameters of the patients.
Response to Comment #2-2
We appreciate this insightful comment. In this study, we did not collect sufficient data related to frailty, making it difficult to add information in this regard. We added this point as a limitation (Page 12, lines 6-8). Both age and BMI are known to be associated with frailty. Therefore, we investigated the relationship between cholesterol metabolism markers and each of age and BMI using different path models in supplementary figure. The path analysis revealed that neither age nor BMI was associated with cholesterol absorption/synthesis. We addressed these points in Page 9, lines 15-31 in the revised manuscript.
Round 2
Reviewer 1 Report
I have no additional comment.